# Alkaline Phosphatase Relieves Colitis in Obese Mice Subjected to Forced Exercise via Its Anti-Inflammatory and Intestinal Microbiota-Shaping Properties

**DOI:** 10.3390/ijms25020703

**Published:** 2024-01-05

**Authors:** Dagmara Wojcik-Grzybek, Zbigniew Sliwowski, Slawomir Kwiecien, Grzegorz Ginter, Marcin Surmiak, Magdalena Hubalewska-Mazgaj, Anna Chmura, Adrianna Wojcik, Tomasz Kosciolek, Aleksandra Danielak, Aneta Targosz, Malgorzata Strzalka, Urszula Szczyrk, Agata Ptak-Belowska, Marcin Magierowski, Jan Bilski, Tomasz Brzozowski

**Affiliations:** 1Department of Physiology, Faculty of Medicine, Jagiellonian University Medical College, 31-531 Cracow, Poland; dagmara1.wojcik@uj.edu.pl (D.W.-G.); slawomir.kwiecien@uj.edu.pl (S.K.); grzegorz.ginter@uj.edu.pl (G.G.);; 2Department of Internal Medicine, Faculty of Medicine, Jagiellonian University Medical College, 31-008 Cracow, Poland; 3Malopolska Centre of Biotechnology, Jagiellonian University, 30-387 Cracow, Poland; 4Department of Biomechanics and Kinesiology, Chair of Biomedical Sciences, Faculty of Health Sciences, Jagiellonian University Medical College, 31-008 Cracow, Poland; jan.bilski@uj.edu.pl

**Keywords:** intestinal alkaline phosphatase, physical exercise, experimental colitis, obesity, adipokines, leptin, inflammation, oxidative stress, microbiome

## Abstract

Intestinal alkaline phosphatase (IAP) is an enzyme that plays a protective role in the gut. This study investigated the effect of IAP treatment on experimental colitis in mice subjected to forced exercise on a high-fat diet. C57BL/6 mice with TNBS colitis were fed a high-fat diet and subjected to forced treadmill exercise with or without IAP treatment. Disease activity, oxidative stress, inflammatory cytokines, and gut microbiota were assessed. Forced exercise exacerbated colitis in obese mice, as evidenced by increased disease activity index (DAI), oxidative stress markers, and proinflammatory adipokines and cytokines. IAP treatment significantly reduced these effects and promoted the expression of barrier proteins in the colonic mucosa. Additionally, IAP treatment altered the gut microbiota composition, favoring beneficial *Verrucomicrobiota* and reducing pathogenic *Clostridia* and *Odoribacter.* IAP treatment ameliorates the worsening effect of forced exercise on murine colitis by attenuating oxidative stress, downregulating proinflammatory biomarkers, and modulating the gut microbiota. IAP warrants further investigation as a potential therapeutic strategy for ulcerative colitis.

## 1. Introduction

Inflammatory bowel disease (IBD) is a group of gastrointestinal disorders characterized by a cyclical nature that alternates between active and quiescent states [1,2]. The two main phenotypic forms of IBDs are Crohn’s disease (CD) and ulcerative colitis (UC) [3]. The high activity of the inflammatory process, systemic complications, the not always good compliance, and patients’ adherence to the applied treatments make these diseases a serious problem. Taking all this into account, IBD significantly deteriorates the physical functioning and quality of life of patients and can even lead to significant disability, creating huge costs for national health care systems [4].

Most IBD patients are underweight, and they are usually malnourished by intense diarrhea, but it is interesting to note that the proportion of intra-abdominal fat is greater than in healthy individuals [5,6]. It turns out that IBDs are characterized by the occurrence of mesenteric white adipose tissue (mWAT) [3]. Visceral fat constitutes the major source of proinflammatory adipokines. In IBDs, the hypertrophic mWAT, so-called creeping fat, seems to be the main factor contributing to the increase in the number of circulating proinflammatory cytokines and may play an important role in the pathogenesis and course of the disease [7]. As hypertrophic mWAT can produce proinflammatory substances leading to disease exacerbation and related complications, working skeletal muscles may secrete anti-inflammatory compounds called myokines that improve the blood supply to organs and mitigate the effects of chronic inflammation.

Some myokines such as irisin can also exert a direct effect on adipose tissue, especially abdominal fat, causing its atrophy. Exercise has been proposed as one of the most important lifestyle practices as a treatment option for IBDs due to the improvements in bone mineral density, fatigue, and quality of life [8]. The mechanism of moderate voluntary physical activity in alleviating experimental colitis in mice has already been studied by our group [9,10]. We have reported that voluntary physical activity can attenuate the severity of colonic damage in obese mice [9,10].

While exercise undeniably offers preventive and therapeutic benefits for chronic diseases, strenuous endurance exercise may worsen existing gastrointestinal disorders [11]. Even healthy individuals can experience digestive symptoms during exhaustive exercise, while participants in demanding events such as marathons may develop “ischemic colitis”, characterized by bloody diarrhea, fatigue, and fever [11,12,13]. Recently, we reported that forced exercise intensified the inflammatory changes in the colon of mice with colitis due to a decrease in blood microcirculation in the colon, an increase in oxidative stress and a rise in the expression and activity of proinflammatory biomarkers [14]. This implies that chronic stress, e.g., by means of repetitive episodes of forced exercise on a treadmill, can exert detrimental effects on physical health, including the progression of human IBD and the course of experimental colitis [15]; however, the mechanism of these harmful changes and the therapeutic options have not been explored.

Several factors, such as changes in the intestinal microbiome, cytokines, and oxidative stress, as well as the dysfunction of mucosal immune system, were recently implicated in the pathogenesis of IBD and mechanism of experimental colitis in rodents [9,10,16,17,18]. Recently, we reported that the severity of experimental colitis is profoundly reduced in obese voluntary exercising mice simultaneously co-treated with an enzyme intestinal alkaline phosphatase (IAP) [10], but the efficacy of this IAP treatment in obese mice fed a high-fat diet and subjected to more intense, i.e., forced exercise, has not been explored before. Alkaline phosphatases are enzymes that catalyze the breakdown of monophosphate esters by hydrolytic phosphate removal [19]. As a result, IAP dephosphorylates toxic microbial cytotoxin, such as lipopolysaccharides (LPS), which are released from cells walls during stressful events [19]. Moreover, this enzyme helps intestinal epithelial cells to form a barrier that prevents the translocation of bacteria and is a major regulator of gut intestinal permeability [20,21]. IAP is then considered one of the most important factors in the protection of the intestinal mucosa; however, whether treatment with IAP could alleviate colitis in obese mice subjected to forced treadmill physical activity has not been studied. The barrier function of the gastrointestinal tract is maintained by the family of tight junction proteins [22] and changes in the expression of tight junctions protein have been correlated with IBDs [23,24,25]; however, to what extent IAP treatment in exercising obese mice with colitis could influence the expression of tight junction proteins remains unknown.

Few human studies have explored the potential of IAP in treating intestinal disorders. Lukas et al. [26] found a short-term improvement in disease activity scores in UC patients after 7 days of IAP treatment with reduced C-reactive protein and stool calprotectin levels. The treatment was well-tolerated and without adverse effects [26]. IBD patients exhibit a decrease in the expression and activity of endogenous IAP [27,28]. Compared to non-inflamed tissues, inflamed colonic tissues from patients with CD and UC demonstrate lower levels of IAP mRNA expression [27,29]. Interestingly, both inflamed and non-inflamed colonic tissues from IBD patients lack the capacity for LPS dephosphorylation in vitro [29].

IAP has potential in clinical settings, especially in treating IBD, reducing exercise-induced gastrointestinal (GI) side effects in IBD patients, and providing protection during strenuous exercise in healthy individuals. IAP supplementation could alleviate these side effects by reducing inflammation and oxidative stress, maintaining the integrity of the gut barrier during strenuous exercise, and reducing the risk of GI distress [10].

The present study was designed to evaluate the effects and mechanism of action of IAP administration in combination with a forced treadmill exercise on experimental colitis in normal mice fed a standard diet (SD) and mice fed high-fat diet (HFD) to induce obesity. We examined the changes in the mice muscle tension as assessed by grip strength test, the macroscopic and microscopic appearance of colonic mucosa, the alterations in intestinal microbiome, oxidative stress biomarkers, and proinflammatory cytokines, and we analyzed the changes in mRNA expression of junctional proteins ZO-1, claudin-4 and claudin-8 in the colonic mucosa of obese mice and in those fed a standard diet.

## 2. Results

### 2.1. Effect of IAP Treatment on Disease Activity Index (DAI), Body Weight and Relative Grip Strength in High-Fat Diet (HFD)- and Standard Diet (SD)-Fed Mice with Trinitrobenzene Sulfonic Acid (TNBS)-Induced Colitis Subjected to Forced Exercise on a Treadmill (T)

Figure 1 shows that that DAI in HFD-fed mice with TNBS colitis forced to participate in treadmill exercise was significantly increased (*p* < 0.05) when compared to SD-fed mice with TNBS colitis forced to participate in treadmill exercise. The IAP administration significantly reduced DAI score in HFD exercising mice with colitis (*p* < 0.05, Figure 1) while it had no significant influence on DAI in exercising animals with colitis fed SD. The IAP did not alter the body weight in vehicle-control mice fed SD; however, the body weight significantly increased in groups of mice with colitis fed an HFD with or without treatment with IAP as compared with SD-fed animals (*p* < 0.05, Figure 1).

Relative grip strength was significantly decreased in HFD-fed mice with colitis forced to exercise on a treadmill when compared to SD-fed mice with colitis forced to participate in treadmill exercise; however, the IAP administration did not change the decrease in the values of relative grip strength as compared with vehicle-treated mice fed an HFD (*p* < 0.05, Figure 1).

### 2.2. The Effect of Forced Treadmill Exercise Combined with IAP Administration on the Macroscopic and Microscopic Appearance of Colonic Mucosa in HFD- and SD-Fed Mice with Colitis

Figure 2 shows the representative gross macroscopic (panel A), microscopic (panel B) and PAS staining (C) appearances of the colon and colonic mucosa obtained from exercising mice with colitis fed an SD or HFD with or without vehicle or IAP administration.

The greatest shortening of colon, as well as the presence of bloody infusion and narrowing in the lumen of the colon, was observed in the TNBS colitis group of exercising mice fed an HFD (panel A), while the lengthening of the colon was noticed in both groups of mice with IAP administration. Histology revealed that the colonic mucosa in the TNBS colitis group of exercising mice fed an HFD showed extensive damage, characterized by necrosis, and inflammatory reaction in lamina propria with neutrophil infiltration caused by TNBS administration. Furthermore, in obese mice, we observed enterocyte hypertrophy. In contrast, less severe damage, moderate neutrophil infiltration and the reconstruction of enterocyte architecture was observed in mice with IAP administration (Figure 2, panel B). PAS staining (panel C) revealed thinner mucus and a decreased goblet cell number in HFD obese mice compared to mice fed an SD and a partial restoration of mucus layer and increase in the number of goblet cells in IAP groups.

### 2.3. Effects of Forced Exercise Effort Combined with IAP Administration on the Mucosal Colonic Content of 8-Hydroxy-2′-Deoxyguanosine (8 OHdG), Malondialdehyde and 4-Hydroxynonenal (MDA+4-HNE), Total Glutathione (GSH+GSSG) Concentration and Superoxide Dismutase (SOD) Activity in HFD- or SD-Fed Mice with TNBS Colitis

Figure 3 shows the changes in the concentrations of 8-OHdG, MDA+4-HNE and GSH+GSSG and the alterations in the activity of SOD in the colonic mucosa of exercising TNBS colitis mice fed an SD or an HFD with or without IAP administration. The treatment with IAP significantly reduced the concentration of 8-OHdG compared to the value of this DNA oxidation marker obtained in forced-exercise mice with TNBS colitis (*p* < 0.05, Figure 4). The lowest mean content of this marker was found in the group of obese mice with colitis exposed to treadmill-running combined with IAP treatment (Figure 3). The lowest total glutathione (GSH+GSSG) concentration was found in the colonic mucosa of obese colitis mice and exercising on a treadmill, and this effect was reversed by IAP administration (*p* < 0.05, Figure 3). In turn, IAP treatment significantly decreased SOD activity as compared to the value observed in non-obese treadmill-running mice with colitis, and this value reached a significantly lower level than that recorded in control mice without IAP administration (*p* < 0.05, Figure 3). The content of lipid peroxidation products was significantly increased in treadmill-exercising obese mice as compared with corresponding SD-fed animals. The concurrent treatment with IAP significantly decreased the colonic level of MDA+4-HNE in obese mice (*p* < 0.05, Figure 3).

### 2.4. Plasma Levels of Pro- and Anti-Inflammatory Cytokines in Exercising TNBS Colitis Mice Fed SD or HFD with or without IAP Treatment

As shown in Figure 4, the concentration of IL-2 was significantly lower in obese mice without IAP administration than in the corresponding group of SD mice. The supplementation of IAP in obese mice significantly increased the IL-2 concentration compared to other groups (*p* < 0.05, Figure 4). In contrast, the IL-6 concentration reached the highest value in group of TNBS colitis mice fed an HFD with treadmill exercise and the treatment with IAP significantly reduced the plasma IL-6 concentration (*p* < 0.05, Figure 4). The plasma level of IL-12p70 tended to increase in mice fed an HDF compared with SD-fed animals but the treatment with IAP failed to significantly influence the concentration of this cytokine (Figure 5). IL-17a had the highest level in the group of obese TNBS colitis mice without IAP administration and treatment with IAP significantly decreased (*p* < 0.05) the plasma levels of this cytokine (Figure 4). The plasma leptin level was significantly higher in both groups of obese animals as compared with SD-fed mice (*p* < 0.05) and the concomitant treatment with IAP significantly reduced the concentration of this hormone as compared that achieved by an HFD without IAP administration (*p* < 0.05, Figure 4). The plasma levels of MCP and TNF-α were significantly higher in obese TNBS colitis treadmill-exercising mice when compared to corresponding SD-fed animals and the administration of IAP significantly decreased the concentration of these cytokines in obese mice (*p* < 0.05, Figure 4).

### 2.5. The Alterations in mRNA Expression of Intestinal Barrier Proteins in Colonic Mucosa of Obese and Normal-Weight Exercising Mice with Experimental Colitis Administered with IAP

Figure 5 shows the alterations in the mRNA expression of tight junction molecules ZO-1, Muc2, Cldn4 and Cldn8 in TNBS colitis mice fed an SD or an HFD subjected to forced exercise with or without the administration of IAP.

In the HFD group, a significant downregulation of mRNA expression for ZO-1, Muc2, Cldn4 and Cldn8 was observed when compared to the SD group (*p* < 0.05, Figure 5). The mRNA expression of ZO-1, Cldn4 and Cldn8 was significantly increased while the Muc2 mRNA expression was significantly decreased in the IAP-treated forced-exercise TNBS colitis rats fed an SD (*p* < 0.05, Figure 5). A significant increase in mRNA expression for ZO-1 and Cldn8 and significant decrease in the expression of Cldn4 and Muc2 mRNAs were observed in the group of obese animals with colitis forced to participate in treadmill exercise (*p* < 0.05, Figure 5). Concomitant treatment with IAP significantly increased the expression of mRNA for intestinal junction proteins in obese mice with colitis when compared to the respective group of obese animals without IAP administration (*p* < 0.05, Figure 5).

### 2.6. Changes in Gut Microbiome of Treadmill-Exercising Mice with TNBS Colitis Fed an SD or HFD with or without IAP Administration

Figure 6 shows the relative abundance of the gut microbiome at the phylum level in treadmill-running obese mice with TNBS-induced colitis. The NGS analysis of the revealed the dominance of *Bacteroidetes* and *Firmicutes* in both studied groups (N = 2). The frequency was 91.52% in obese mice without IAP administration and 81.69% in mice with the IAP intervention. The relative abundance of *Verrucomicrobiota* reached 10.83%, and these bacteria were most prevalent in the group of mice treated with IAP compared with another group. Both *Campylobacteria* and *Deferribacterota* were more common in group HFD + T + TNBS (2.72% and 2.66%, respectively, vs. 1.82% and 1.96%, respectively, in HFD + T + IAP + TNBS). *Actinobacteria* and *Proteobacteria* did not exceed 1% in mice without IAP administration while Proteobacteria exceeded 2% in mice with IAP. *Desulfobacterota’s* relative abundance in obese, treadmill-running, TNBS colitis mice was 1.31%, while in mice with IAP treatment combined with treadmill exercise, it reached the value of 0.93% (Figure 6).

We also investigated microbiome composition at the genus level for the same groups of mice (Figure 7). In both groups, the dominance of *Muribaculaceae* and *Alloprevotella* was noticed. Interestingly, the *Odoribacter* was the next most abundant group in the HFD + T + TNBS group, while *Alistipes’* relative abundance reached a similar level in both groups ~6.3% (Figure 7). Interestingly, Clostridia was detected only in the group of treadmill exercising TNBS colitis mice without combination with IAP, and, similarly, *Oscillispiraceae* and *Lachnospiraceae* were also more abundant in the group without IAP treatment.

## 3. Discussion

The present study demonstrates that experimental colitis was alleviated by administering IAP in obese mice; apparently, these mice became stressed by being forced to exercise on a treadmill. This beneficial effect of IAP was associated with a reduction in the level of proinflammatory biomarkers in plasma, attenuation of oxidative stress markers in the colonic mucosa, favorable shift in the intestinal microbiota limiting the abundance of pathogenic microorganisms, and an improvement in the intestinal barrier tight junction’s proteins, reflecting an increase in the gene expression of the junctional proteins ZO-1, Muc2, Cldn4 and Cldn8. In the mice fed a standard diet, these changes were not as pronounced, despite some similar tendencies to those monitored in obesity. Thus, our present study sheds more light on the mechanism of action of exogenously administered IAP, which exhibited a protective effect in ameliorating the severity of experimental colitis in mice with diet-induced obesity who were forced to exercise.

To our best knowledge, herein, we present, for the first time, that the administration of IAP reduced the negative effects of forced treadmill-running in mice with colitis. In our previous study, we documented that forced running on a treadmill, unlike voluntary exercise, increased colon damage in obese mice [9,10,14]. This study confirmed our previous observation [16] that obesity elevates DAI score in mice with experimental colitis forced to participate in physical exercise. The gross macroscopic and microscopic assessments revealed the shortening of the colon with the enterocyte hypertrophy of colonic mucosa in control obese, exercising mice with colitis. Moreover, our present data document that IAP administration significantly lowered DAI, as reflected by the reduction in mucosal bleeding and the severity of damage due to hemorrhagic lesions, as well as the reconstruction of normal enterocyte architecture in these animals. Recently, Hwang et al. demonstrated that the oral administration of IAP significantly alleviated the severity of murine colitis, mainly by reducing the production of the anti-inflammatory cytokines TNF-α and IL-6 [30]. Our current data, along with a previous report [10], remain in agreement with this observation, since a major improvement in colitis was predominantly observed in obese mice. Interestingly, Hwang et al. [30] have observed a reduction in the body weight of mice treated with IAP, an effect apparently not observed in our present study. The reason for this discrepancy can be explained by methodological differences between the previous report by Hwang et al. [30] and our present study, related to the selection of a different colitis model in both studies, namely DSS in their study [30] vs. the single intrarectal administration of TNBS employed by us. Moreover, they supplemented mice with IAP during colitis development [30] while, in our present study, IAP was administered prior to colitis induction. Nevertheless, we have also shown an improvement in the gross and histological appearance of the colon in mice treated with IAP, suggesting that IAP treatment may deserve further clinical interest as a possible therapeutic option in IBD. We have shown that the relative grip strength is lower in obese forced-exercise mice; however, the concurrent administration of IAP in treadmill-exercising mice with colitis did not change relative grip strength. This observation is corroborated by the results on this parameter in mice exercising voluntarily on spinning wheels [10].

Our observations regarding the beneficial effects of IAP are supported by earlier reports that a reduced expression of IAP was found in IBD, metabolic syndrome, cystic fibrosis, necrotizing enterocolitis, and diabetes [19,28,31,32], and IAP deficiency has been linked with obesity and IBD [10,28,33,34]. Moreover, this enzyme has been included as essential in the mechanism of normalizing of the homeostasis of the intestinal microbiota [32]. That is why the aim of our study was to examine changes in the intestinal microbiota associated with the effect of treadmill exercise on TNBS colitis. Our analysis of these changes mainly considered obese mice due to the expectation of a greater improvement in the inflammatory and oxidative stress indices caused by IAP as compared with these parameters in groups without IAP treatment.

The NGS analysis of the intestinal microbiome in our present study revealed the dominance of *Bacteroidetes* and *Firmicutes* in both studied groups at the *phylum* level, confirming previous results [10,35,36]. Laukens et al. [36] and Turnbaugh et al. [35] identified *Bacteroidetes* in 20–40% of the studied sequences and 60–80% of the *Firmicutes* sequences in mice. Our previous study [10] found higher levels of *Bacteroidetes* (around 50%) and a lower level of *Firmicutes* (around 45%) in groups of obese mice with voluntary exercise and/or IAP administration, suggesting that microbiota might be influenced by the intensity of exercise. It has been shown that the intestinal presence of *Bacteroidetes* in the lower gut is directly related to body weight and the relative proportion of *Bacteroidetes* increases proportionally with the body weight of obese humans and animal models with diet-induced obesity [37,38], in keeping with the results of in our present study. Moreover, the intestinal abundance of *Bacteroidetes* can reach around 60% in IBD patients. The group administered with IAP showed a smaller number of *Firmicutes*, in favor of *Verrucomicrobiota*, increasing the diversity of microbiota. *Verrucomicrobiota* is a part of the normal gut microflora and rarely increases during active phases of intestinal mucosal inflammation, but can be found in IBD patients under remission [39,40,41]. Due to the abundance of Verrucomicrobiota, Akkermansia and Lactobacillus, which have been considered beneficial bacteria in IBD [41], our present study implies that IAP improved intestinal inflammation due to the repair of gut dysbiosis and reduction in intestinal mucosal permeability in obese mice with colitis. Thus, IAP appears to help restore the physiological gut microbiota and bacteria diversity.

At the genera level, *Muribaculaceae*, *Turicibacter* and *Lachnospiraceae* have previously been identified as key targets for microbiota alterations in colitis. Among others, *Helicobacter*, *Mucispirillum*, *Clostridiales_vadinBB60_group* and *Odoribacter* were positively correlated with inflammation-related parameters [42]. In our study, the four abovementioned genera were more abundant in the group without IAP administration, which supports our notion that this enzyme reduced the content of negative, perhaps pathogenic, microbiota. Furthermore, *Muribaculaceae* was more numerous in the group of mice with IAP supplementation. This implies that, in the presence of IAP, the gut microbiota changes from a deleterious state of pathogenicity to normal commensal flora.

The integrity of the gut barrier can prevent exogenous substances such as LPS from entering the bloodstream and reducing oxidative stress and inflammation. The intestinal barrier plays a key role in human health and disease. The disruption of intestinal integrity resulting in a “leaky gut” causes penetration by luminal inflammatory mediators including LPS into the subepithelial environment and the systemic blood circulation. This process leads to harmful immune responses and inflammation in various organs. As a result, impaired intestinal mucosal barrier function is implied to play an important role in intestinal diseases such as IBD [43,44]. A ‘leaky gut’ is considered as a common complication in IBD at present [45]. An important element of epithelial barrier function is the maintenance of intercellular tight junctions, which are closely related to the antioxidant and anti-inflammation effects and consist of proteins including occludin, claudins and zonula occludens-1 (ZO-1). Liu et al. [21] have shown that IAP is the primary permeability regulator of the intestinal mucosa and appears to work by improving the structure and function of tight junction proteins. This enzyme exhibits a protective effect against inflammation through the detoxification of bacterial LPS, regulation of gut microbiome, dephosphorylation of proinflammatory nucleotides, regulation of bicarbonate secretion and pH of the duodenal surface, as well as the absorption of intestinal long-chain fatty acids [46].

Our previous observations showed that treatment with exogenous IAP can improve inflammation, possibly by enhancing gut barrier function and integrity in colitis in obese mice [10]. This is why we examined changes in the mRNA expression of junction proteins ZO-1, claudins (Cldn4 and 8) and mucin 2 (Muc2). For instance, Muc2 forms the backbone of the intestinal mucus lining the intestinal epithelium. Our current findings provide new insight into mechanism of action of IAP, demonstrating that treatment with exogenous IAP enhanced the expression of crucial tight junction proteins, thus enhancing gut barrier integrity and its function in obese mice with colitis, an effect contributing to the overall improvement in the resolution of intestinal inflammation. The Muc2 forms the skeleton of the mucus, which covers and protects the digestive tract from self-digestion and prevents the harmful action of pathogenic microorganisms in various segments of the GI-tract [47]. The intestinal microbiota remains in close contact with mucin and antimicrobial peptides that, in turn, are involved in the mechanism of regulation of the microbial constituents [15]. Our results showed a significant decrease in Muc2 mRNA in obese colitis mice, an effect reversed in mice administered with IAP, suggesting, on the one hand, a possible reduction in mucus production capacity in obesity and, on the other hand, supporting the idea that IAP contributes to the restoration of mucus formation. In fact, this notion is supported by our observation that, in obese mice, thinner mucus and a reduced number of goblet cells were observed, as assessed by PAS staining, compared to control mice kept on a standard diet. This restoration of mucus layer thickness and increase in goblet cell numbers were achieved when IAP was administered to forced-exercise mice with colitis. These findings are corroborated by the observation in the rat model of TNBS-induced colitis that the expression of Muc2 is downregulated [48]. Gao et al. [15] found no significant differences in Muc2 expression in murine colitis; however, this effect may be related to the method of colitis induction, namely the use of a chronic, less severe model of DSS colitis that may readily accelerate a faster defense response, leading to mucus restoration.

ZO-1 is considered a protein by which claudins directly bind to the central scaffold [25]. In our study, ZO-1 mRNA expression was significantly higher in both groups treated with IAP. Previous studies in human IBD, mice with DSS colitis, and rats with TNBS colitis have shown the downregulation of ZO-1 expression compared to healthy controls, confirming that in both human IBD and the course of experimental colitis, the ZO-1 is downregulated [25,48,49,50,51]. Our study is the first to confirm that IAP counteracts this deleterious effect by increasing ZO-1 expression, possibly reducing colonic permeability in obese colitis mice. We also assessed the expression of claudins Cldn4 and Cldn8 in the colonic mucosa since the expression of Cldn4 has previously been shown to be downregulated in patients with ulcerative colitis [52]. Interestingly, human claudin-4 was overexpressed in colorectal cancer (CRC), suggesting that this barrier protein may be a potential marker of CRC [53]. In addition, the expression of proteins Cldn4 and Cldn8 has been shown to decrease in lymphocytic colitis [54]. We observed that Cldn4 was downregulated in exercising obese mice without IAP treatment when compared to mice fed an SD. Our study revealed that, in IAP-treated mice with colitis, both Cldn4 and Cldn8 were upregulated. This could be convincing evidence that IAP may affect the gut permeability by regulating the mRNA expression of tight junction proteins under inflammatory conditions.

Our current study has shown that the protective effect of IAP against the forced-exercise-induced aggravation of colitis is associated with the fall in the plasma levels of proinflammatory biomarkers and a marked reduction in indices of oxidative stress. Oxidative stress plays an important role in the pathogenesis of colitis [10,16,55,56,57,58]. During the inflammatory process associated with the development of colitis, the cells of the immune system (macrophages, neutrophils, and lymphocytes) invade the inflamed mucosal layer and release excessive reactive oxygen species (ROS), as well as proinflammatory cytokines. Moreover, long-standing inflammation caused by oxidative stress is the most important risk factor for CRC development, and the determination of 8-OHdG is widely accepted as the most common marker of DNA damage [59,60].

We found that the administration of IAP reduced the content of 8-OHdG in normal-diet-fed mice with colitis forced to exercise on a treadmill. The concentration of this marker remained unchanged in both groups with IAP administration, indicating that this protective IAP intervention is a promising therapeutic option against the development of DNA damage, thereby accounting for an improvement in the healing of colitis in our study. The level of reduced glutathione is an indicator of the intensity of antioxidant processes in response to oxidative stress and indicates the effectiveness of scavenging ROS [10]. Previous reports have indicated that the lowering of GSH and GSSG content occurs during the course of colitis [48,55]. In this study, an increase in GSH+GSSG concentration was observed after the administration of IAP to only obese mice. Moreover, the administration of IAP ameliorated lipid peroxidation, as reflected by the fall in MDA content. We found that the content of MDA and 4-HNE was increased in exercising obese mice, and this measure was clearly decreased by the treatment with IAP. Therefore, our results indicate that the restoration of reduced glutathione and decrease in lipid peroxidation could explain the beneficial effect of this enzyme administration in obese mice with colitis. Reducing the level of ROS in inflamed tissues has been recognized as an effective strategy to alleviate enteritis during colitis [61]. Bai et al. [62] showed that the administration of an encapsulated SOD antioxidant enzyme can effectively eliminate superoxide anions, thus reducing oxidative damage in the intestinal tissue. Herein, we observed a tendency for the SOD activity to increase in obese animals versus respective controls, and this was significantly decreased in HFD-mice treated with IAP, similar to a recent report by our group [10]. Furthermore, IAP administration reduced the plasma levels of pro- and anti-inflammatory biomarkers in obese mice. For instance, interleukin (IL)-2 is an important anti-inflammatory cytokine that also contributes to immune changes during inflammation and obesity [63]. In this study, the plasma levels of IL-2 were significantly lower in obese mice, and treatment with IAP suddenly increased this level. Moreover, the content of proinflammatory cytokines were also reduced by IAP, indicating the attenuation of inflammatory response in obese mice with colitis subjected to treadmill exercise. Apart from the previously mentioned TNF-α and IL-6, the mean plasma levels of IL-6, leptin, MCP and IL-17a decreased in IAP-treated TNBS colitis obese mice forced to participate in treadmill exercise. These data further explore the notion that IAP regulates the immune response associated with intestinal inflammation, and this serves as an explanation for the resolution of the intensity of the inflammation in obese mice with colitis. In this case, additional stressors, besides obesity, such as forced running on a treadmill, appear to be better tolerated by the animals treated with IAP.

In summary, the present study points to the administration of IAP as an alternative option to pharmacotherapy, as documented in our experimental study in colitis mice stressed by forced exercise on treadmill. We provided an insight into this beneficial effect of IAP by demonstrating that the administration of this protective enzyme reduced pro-inflammatory biomarkers, attenuated biomarkers of oxidative stress, improved the expression of intestinal barrier tight junction proteins, and altered the composition and diversity of intestinal microbiota. It is of interest that IAP attenuated the inflammatory response, particularly in obese mice, possibly due to the reduced expression and release of proinflammatory cytokines, reduced oxidative stress, improved expression of tight junction proteins, and beneficial changes in the gut microbiome towards an abundance and increase in the diversity of normal microbiota (see Graphical Abstract). The results obtained in this study may be of translational importance for the clinic when improving the quality of lives of especially obese patients, who sometimes try to lose weight through endurance or strenuous exercise, and at the same time suffer from IBD in stressful everyday life. Therefore, treatment with IAP exerting a beneficial effect in experimental colitis resolution deserves further attention and is of clinical interest as an interesting option in future clinical trials in IBD patients.

## 4. Materials and Methods

### 4.1. Animals and Diets

Studies were performed on 28 female C57BL/6J mice with initial weight 18–22 g. Mice were 10 weeks old and kept in a pathogen-free cage with constant control of temperature, ventilation, and humidity (Bioscape Bio. A.S., Amsterdam, The Netherlands). Animals had free access to water and food and were kept in laboratory conditions with 14 h/10 h day/night cycles.

The study was approved by the local Ethical Committee at the Jagiellonian University Medical College in Cracow, Poland (No. 19/2016) and was run in accordance with the Helsinki declaration, with implications for replacement, refinement, or reduction (the 3Rs) (Decision No.: 19/2016; date: 20 July 2016).

### 4.2. Experimental Design

Animals were subjected to an adaptation period for 3 days after the purchase. After that time, they were randomly assigned into 4 experimental series, each consisting of 6 animals per group:

(1)Mice kept on a standard diet (SD), subjected to forced exercise on treadmill (T) and with colitis induced by 2,4,6-trinitrobenzenesulfonic acid (TNBS) (SD + T + TNBS);(2)Mice kept on a standard diet, subjected to forced exercise on treadmill, administered with intestinal alkaline phosphatase (IAP) and with induced colitis (SD + T + IAP + TNBS);(3)Mice fed a high-fat diet (HFD), subjected to forced exercise on treadmill and with induced colitis (HFD + T + TNBS).(4)Mice fed a high-fat diet, subjected to forced exercise on treadmill, administered with IAP followed by colitis induction (HFD + T + IAP + TNBS).

Standard diet groups (1 and 2) were fed a regular chow pellet (SD, diet C 1000; Altromin, Lage, Germany). High-fat diet groups (3 and 4) were fed for 12 weeks with a high-fat diet (HFD), diet C 1090-70—obesity-inducing diet with w/70% energy from fat (42% pork fat), as described previously [9,10,16]. All groups of animals were then subjected to forced physical exercise on treadmill (no. 76-0894, Panlab, Harvard Apparatus, MA, USA) for 6 weeks, 5 days per week, with speed 10 cm/s for 10 min. Then, mice from groups 2 and 4 were administered with IAP and from groups 1 and 3 were administered with water. Based on the results of our previous study [10], in which the oral administration of IAP (200 U/day) attenuated chronic colitis in mice, animals were administered with the enzyme at a daily dose of 200 U (P0114, Sigma Aldrich, MO, USA), for 2 weeks, in drinking water. During the time of experimentation, mice were still maintained on an SD or HFD, respectively. After IAP administration, the experimental colitis was induced. After 5 days, animals were then weighted, their grip strength was measured, and their stool samples were collected (Figure 8).

### 4.3. Grip Strength Test

Grip strength was measured using a grip strength test device (BIO—GS3, Pinellas Park, FL, USA) according to the manufacturers’ instructions, as described previously [10]. The measurement was carried out three times, with a few minutes’ interval for each mouse. The arithmetic mean of the three measured values of this test was considered the correct measurement result. The unit of measurement is newton [N]. Relative grip strength is expressed as mean grip strength for each group, divided by mean body weight for each group [N/kg].

### 4.4. Experimental Colitis Induction

The experimental colitis was induced in all groups by the intra-colonic administration of TNBS, as described previously [9,10,16]. In brief, animals were anesthetized by inhalation of isoflurane (2–3% in a breathing mixture with oxygen; Aerrane, PGF, Wroclaw, Poland) using the Ugo Basile compact anesthesiologic system (Ugo Basile No. 21100, Gemonio, Italy). A dose of 4 mg of TNBS (water solution, no. 92822, Sigma Aldrich, Burlington, MA, USA) dissolved in 50% ethanol solution was applied rectally at a volume of 175 μL per mouse, using a soft polyethylene catheter (Instech, Plymouth Meeting, PA, USA), about 4 cm deep. Until awakening, the animals were placed in Trendelenburg position. Animals from the control group received rectally, analogous to TNBS, a volume of 50% ethanol with water. To prevent dehydration, all animals were given a subcutaneous injection of 1 mL of saline.

At day 5 after colitis induction, the animals were sacrificed with i.p. lethal dose of pentobarbital (Biowet, Pulawy, Poland). The abdominal cavity was opened, and the colon was separated. The disease activity index (DAI) was calculated using a modification of a previously published compounded clinical score [10].

In brief, the DAI comprised the scoring for diarrhea and lethargy (0–3) and rectal bleeding assessment involved a visual inspection of blood in feces and the perianal area (0–5). A “0” represents a healthy colon and “5” represents the most intensive course of the disease, with bleeding rectal and extensive ulcers.

Samples of colonic tissue were collected on ice, snap-frozen in liquid nitrogen, and stored at −80 °C until further analysis. Blood was drawn from the *vena cava*. About 200 mg of fecal samples was collected for each group and stored at −80 °C until further analysis by next generation sequencing method.

### 4.5. Macroscopic and Microscopic Changes in the Colonic Mucosa Assessment

For a histological determination of the colonic mucosa, the samples of colonic tissue were excised and fixed in 10% buffered formalin with pH = 7.4. Samples were dehydrated and embedded in paraffin blocks, which were cut into about 4 μm sections using a microtome. The prepared specimens were stained with hematoxylin/eosin (H&E) and evaluated using a light microscope (AxioVert A1, Carl Zeiss, Oberkochen, Germany), as described previously [10]. A digital documentation of histological slides was stored and evaluated using the abovementioned microscope equipped with an automatic scanning table and ZEN Pro 2.3 software (Carl Zeiss, Oberkochen, Germany). For histological assessment, the 10 μm thick sections were scored using a histopathological score for intestinal inflammation in mice proposed by Erben et al. [64]: 1—mild mucosal inflammatory cell infiltrates intact epithelium, 2—inflammatory cell infiltrates into mucosa and submucosa with undamaged epithelium, 3—mucosal infiltrates focal ulceration; 4—inflammatory cell infiltrates in mucosa and submucosa and focal ulceration, 5—moderate inflammatory cell infiltration into mucosa and submucosa with extensive ulcerations; 6—transmural inflammation and extensive ulceration [10,64].

### 4.6. Next Generation Sequencing of Gut Microbiome

DNA was isolated from stool samples using QIAamp DNA Stool Mini Kits (Qiagen, Hilden, Germany) according to the manufacturers’ protocol, as described previously [10]. An evaluation of bacterial population was performed based on the analysis of the hypervariable V3–V4 region of 16S ribosomal RNA (rRNA) [17,65]. For the region amplification and library preparation, specific sequences of 341F and 785R primers were used (16S analysis). PCR was performed using Q5 Hot Start High-Fidelity 2X Master Mix (New England Biolabs, Ipswich, MA, USA) according to the manufacturer’s protocol. DNA sequencing was performed using MiSeq sequencer (Illumina, Inc., San Diego, CA, USA) and paired-end technology. The data were analyzed by MiSeq Reporter (MSR) v2.6 (Illumina, Inc., San Diego, CA, USA). The analysis was based on automatic sample demultiplexing and FASTQ file generation containing raw data. Bioinformatic analysis up to the genus level was performed by the QIIME software (http://qiime.org/1.4.0/# accessed on 28 December 2023) based on the reference database SILVA_v_138 [66,67] (Caporaso et al., 2010; Quast et al., 2013). Additional analyses and graphs were established using R software (https://www.r-project.org/ accessed on 28 December 2023) and ALDEx2 and ggplot2 package (https://www.bioconductor.org/packages/release/bioc/html/ALDEx2.html accessed on 28 December 2023) [68]. Data are presented as a percentage of bacterial taxa in each sample. An alpha-rarefaction analysis was performed, which allows for an estimation of the extent to which we can observe the full diversity of bacteria within each of the samples and how the diversity varies between groups. Two measures of diversity were used: observed features and Shannon entropy. A taxonomic analysis was also performed using the Naive Bayes algorithm and a classifier trained using Silva 99%.

### 4.7. Determination of DNA Oxidation Levels by 8-Hydroxy-2′-Deoxyguanosine (8-OHdG) Concentration

For the measurement of 8-OHdG concentration as a DNA oxidation marker, the DNA was isolated from colonic mucosa using ELISA kit (589320, Cayman Chemical, Ann Arbor, MI, USA) according to the manufacturer’s protocol, as described previously [10,69,70]. In brief, the collected colon fragments were homogenized in phosphate buffer (pH = 7.4) with EDTA. The samples were then centrifuged, and the supernatant was purified using the Tissue DNA Purification Kit (cat. No. E355, EURx, Gdansk, Poland). Total DNA concentration was measured using a Qubit 3 fluorimeter (Thermo Fisher Scientific, Waltham, MA, USA). The obtained DNA was incubated with P1 nuclease and then adjusted to pH 7.5–8.5. Alkaline phosphatase was added, and samples were incubated for 30 min at 37 °C, and then at 100 °C for 10 min, and placed on ice. The test sample, the 8-OH-dG-acetylcholinesterase conjugate and the monoclonal antibody were added to the plate coated with the goat polyclonal antibody, and incubated. After that, the membrane was detached and Ellman’s reagent was added. The absorbance was measured with reagent blank and different concentrations of standards with a microplate reader (Tecan Sunrise, Mannedorf, Switzerland) at a wavelength of 412 nm. The 8-OHdG concentration was expressed as ng of 8-OHdG per 1 µg of total DNA.

### 4.8. Lipid Peroxidation Determination

The concentration of malondialdehyde (MDA) and 4-hydroxynonenal (4-HNE) in colonic samples was measured, as described previously [10,14]. The spectrophotometric method, using the kit for lipid peroxidation (Bioxytech, LPO-586, Oxis, Portland, OR, USA), was used. In brief, the colonic mucosal samples were excised and transferred to the vial containing butylated hydroxytoluene in acetonitrile in PBS (pH = 7.4). Samples were subsequently mechanically homogenized and centrifuged. The obtained clear supernatant was stored at −80 °C until assayed. Two independent biological replicates were performed. The absorbance at 586 nm was analyzed with a microplate reader (Tecan Sunrise, Mannedorf, Switzerland). Results were expressed as nanomoles per gram of colonic tissue (nmol/g).

### 4.9. Total Glutathione (GSH+GSSG) Concentration Measurement

To determine the concentration of total glutathione, the colorimetric assay using an enzymatic recycling method (Cat# 703002, Cayman Chemical, Glutathione Assay Kit, Ann Arbor, MI, USA) was used, as described previously [10]. In short, the colonic samples were collected and homogenized in MES, pH 6–7, containing EDTA. The homogenates were centrifuged, and the upper clear aqueous layer was collected and deproteinated. The obtained supernatant was collected, and the TEAM Reagent was added. Two independent biological replicates were performed. The level of total glutathione was measured with maximal absorbance at 410 nm by a microplate reader (Tecan Sunrise, Mannedorf, Switzerland). Results were expressed as nanomoles per gram of tissue (nmol/g).

### 4.10. Superoxide Dismutase (SOD) Activity Determination

For the measurement of the SOD activity, the colorimetric assay was used (Cat# 706002, Cayman Chemical, Ann Arbor, MI, USA) as described previously [10,14]. In brief, samples of colonic mucosa were collected and homogenized in cold HEPES buffer (pH = 7.2) and centrifuged. The supernatant was collected and immediately assayed. The absorbance was measured using microplate reader (Tecan Sunrise, Mannedorf, Switzerland) at 450 nm. The results were expressed as units per gram of colonic tissue (U/g).

### 4.11. Luminex Microbeads Fluorescent Assay and mRNA gene expression of intestinal barrier proteins ZO-1, MUC-2, Cldn-4 and Cldn-8

The determination of plasma IL-2, IL-6, IL-12p70, IL-17a, TNF-α, MCP and leptin levels was performed using Luminex microbeads fluorescent assay (Bio-Plex Pro™, Hercules, CA, USA Mouse Cytokine) and Luminex 200 system (Luminex Corp., Austin, TX, USA). Results were calculated from calibration curves and expressed in pg/mL of plasma blood for IL-2, IL-6, IL-10, IL-12p70, IL-17a, TNF-α, MCP-1 and in ng/mL of plasma blood for leptin, as described in detail previously [9,10].

An analysis of mRNA expression in colonic mucosa was conducted using real-time polymerase chain reaction (PCR).

Total cellular RNA was isolated from mouse colonic mucosal tissue according to the Chomczynski and Sacchi method [71] using Trizol Reagent (Invitrogen, Carlsbad, CA, USA). First-strand cDNA was synthesized from total cellular RNA (2 µg) using Reverse Transcription System (Promega, Madison, WI, USA). The PCR was carried out from tissue using 1 µg cDNA and Promega PCR reagents. Specific primers for β actin, ZO-1, MUC-2, Cldn-4 and Cldn-8 were used (Sigma-Aldrich, St. Louis, MO, USA) [72]. Sequences and annealing temperatures are listed in Table 1.

PCR products were separated by electrophoresis in 2% agarose gel containing 0.5 µg/mL ethidium bromide and then visualized under UV light. The location of the predicted PCR product was confirmed using O’Gene Ruler 50 bp DNA ladder (ThermoFisher, Waltham, MA, USA) as standard marker. Densitometric analysis was performed using Image Studio Lite program (LI-COR Biotechnology, Lincoln, NE, USA).

### 4.12. Statistical Analysis

Results are expressed as means ± SEM. The data were processed by the GraphPad Prism 5.0 software (GraphPad Software Inc., La Jolla, CA, USA), except the microbiome analysis described in detail above. Statistical analyses were conducted using Student’s *t*-test or ANOVA with Dunnett’s multiple comparison, or Tukey’s post hoc test if more than two experimental groups were compared. The size of each experimental group was of n = 6–8. Type I statistical error *p* < 0.05 was considered significant.

## 5. Conclusions

The present study provides compelling evidence supporting the therapeutic potential of IAP in alleviating experimental colitis in obese mice subjected to forced treadmill exercise. The administration of IAP was associated with a significant reduction in pro-inflammatory biomarkers, an attenuation of oxidative stress markers, an improvement in intestinal barrier tight junction proteins, and beneficial alterations in the gut microbiota. A particularly noteworthy observation was the significant reduction in plasma leptin levels following IAP treatment in obese mice with colitis. The observed decrease in leptin levels after IAP administration suggests that IAP may not only mitigate inflammation but could also potentially contribute to weight management in the context of obesity. Furthermore, the study highlights the promise of IAP as a therapeutic option for IBD, particularly in obese patients who would like to urgently lose weight. The findings suggest that IAP treatment could improve the quality of life of these patients by reducing the severity of colitis and mitigating the negative effects of excessive physical exercise. Although further research is needed to fully elucidate the mechanisms underlying the beneficial effects of IAP and to validate these findings in clinical settings, the current study provides a strong foundation for further investigation of IAP as a novel therapeutic approach for IBD.

## Figures and Tables

**Figure 1 ijms-25-00703-f001:**
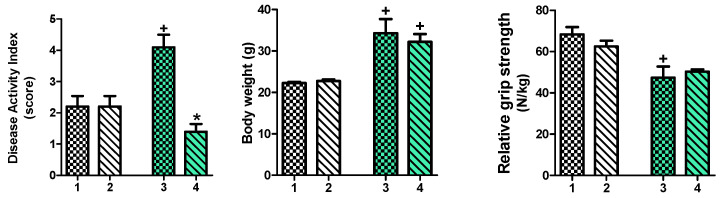
The changes in DAI score, body weight (in grams) and relative skeletal muscle grip strength (expressed in Newton (N) per kilogram of body weight) of mice with colitis fed an SD or an HFD and forced to partake in treadmill exercise, with or without IAP administration. Results are mean ± S.E.M. of 6–8 animals per each group. A cross indicates a significant change (*p* < 0.05) as compared to the respective values in corresponding SD-fed mice. An asterisk indicates a significant change (*p* < 0.05) as compared to the respective values in the HFD exercising mice with colitis. 1. Standard diet (SD) + Treadmill (T) + TNBS. 2. SD + T + IAP + TNBS. 3. High-fat diet (HFD) + T + TNBS. 4. HFD + T + IAP + TNBS.

**Figure 2 ijms-25-00703-f002:**
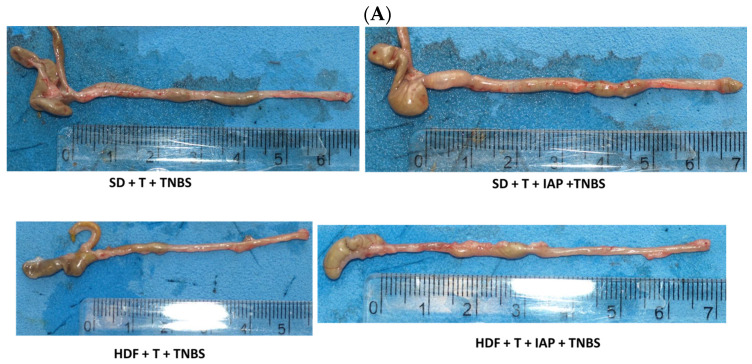
(**A**–**C**) The representative gross appearance of the colon (panel (**A**)) and microscopic appearances of colonic mucosa (panel (**B**), H&E staining) and (panel (**C**), PAS staining) obtained from TNBS colitis mice fed an SD or HFD and forced to participate in treadmill (T) running with or without IAP administration as assessed at day 5 from colitis induction.

**Figure 3 ijms-25-00703-f003:**
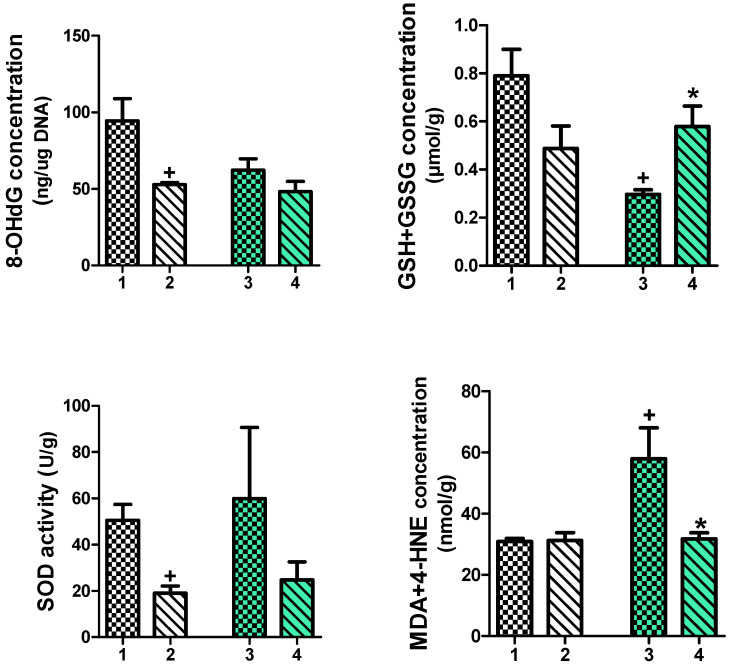
The changes in DNA oxidative damage, expressed as 8-OhdG content; the total glutathione, expressed as GSH+GSSG concentration; the superoxide dismutase (SOD) activity; and the changes in lipid peroxidation products, expressed as the MDA+4-HNE concentration, in the colonic mucosa of treadmill-exercise colitis mice fed an SD or HFD with or without IAP administration. Results are mean ± S.E.M. of 6–8 animals per group. A cross indicates a significant change (*p* < 0.05) as compared to the respective values in the TNBS mice fed an SD with treadmill effort. An asterisk indicates a significant change as compared to the respective values in TNBS colitis mice fed an HFD without IAP administration (*p* < 0.05). 1. Standard diet (SD) + Treadmill (T) + TNBS. 2. SD + T + IAP + TNBS. 3. High-fat diet (HFD) + T + TNBS. 4. HFD + T + IAP + TNBS.

**Figure 4 ijms-25-00703-f004:**
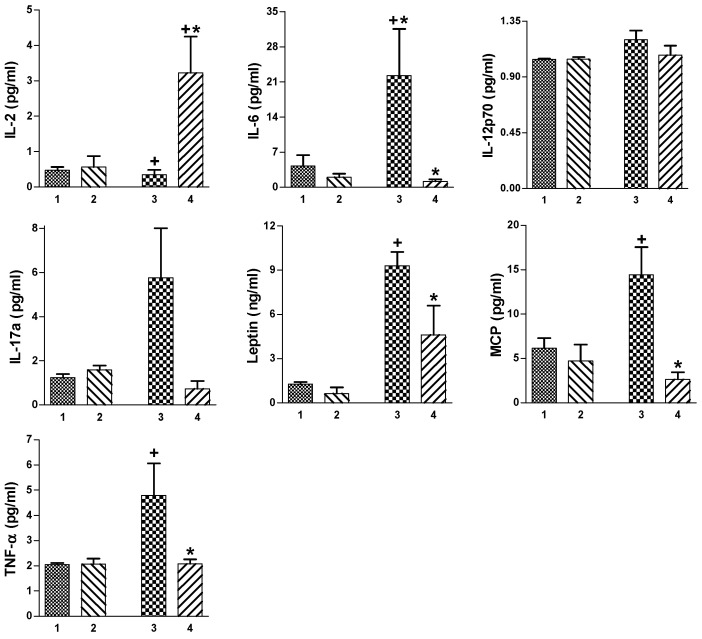
Plasma level of IL-2, IL-6, IL-12p70, IL-17a, leptin, MCP and TNF-α in TNBS colitis mice fed an SD or HFD and forced to participate in treadmill exercise (T) with or without IAP administration. Results are mean ± S.E.M. of 6–8 animals per group. A cross indicates a significant change (*p* < 0.05) as compared to the respective values in the corresponding SD group: SD + T + TNBS for SD + T + IAP + TNBS and HFD + T + TNBS for HFD + T + IAP + TNBS. An asterisk indicates a significant change as compared to the respective values in the HFD + T + TNBS group. 1. Standard diet (SD) + Treadmill (T) + TNBS. 2. SD + T + IAP + TNBS. 3. High-fat diet (HFD) + T + TNBS. 4. HFD + T + IAP + TNBS.

**Figure 5 ijms-25-00703-f005:**
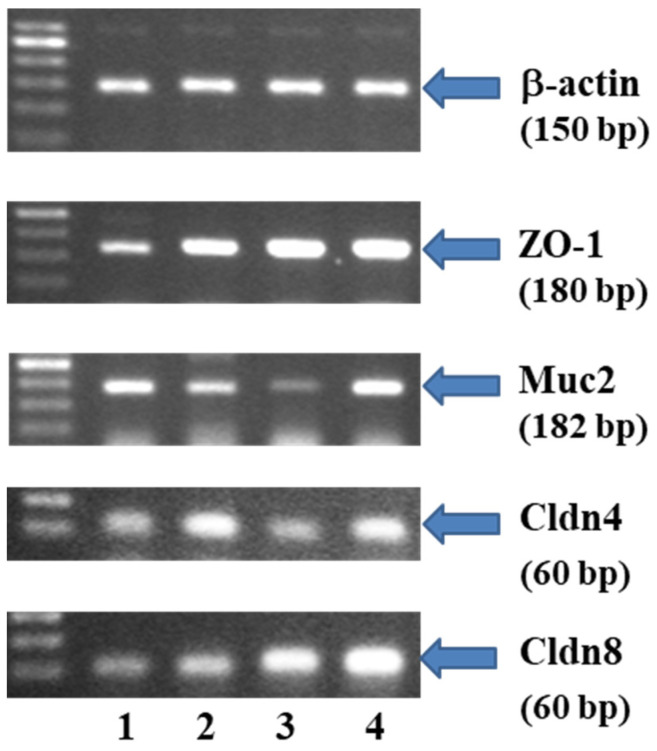
The alterations in the mRNA expression of intestinal barrier proteins ZO-1, Muc2 and claudins (Cldn4 and Cldn8) in colonic mucosa of mice with TNBS colitis fed an SD or HFD and exposed to forced treadmill exercise with or without IAP administration. Results are mean ± SD of 6–8 animals per group. 1. Standard diet (SD) + Treadmill (T) + TNBS. 2. SD + T + IAP + TNBS. 3. High-fat diet (HFD) + T + TNBS. 4. HFD + T + IAP + TNBS.

**Figure 6 ijms-25-00703-f006:**
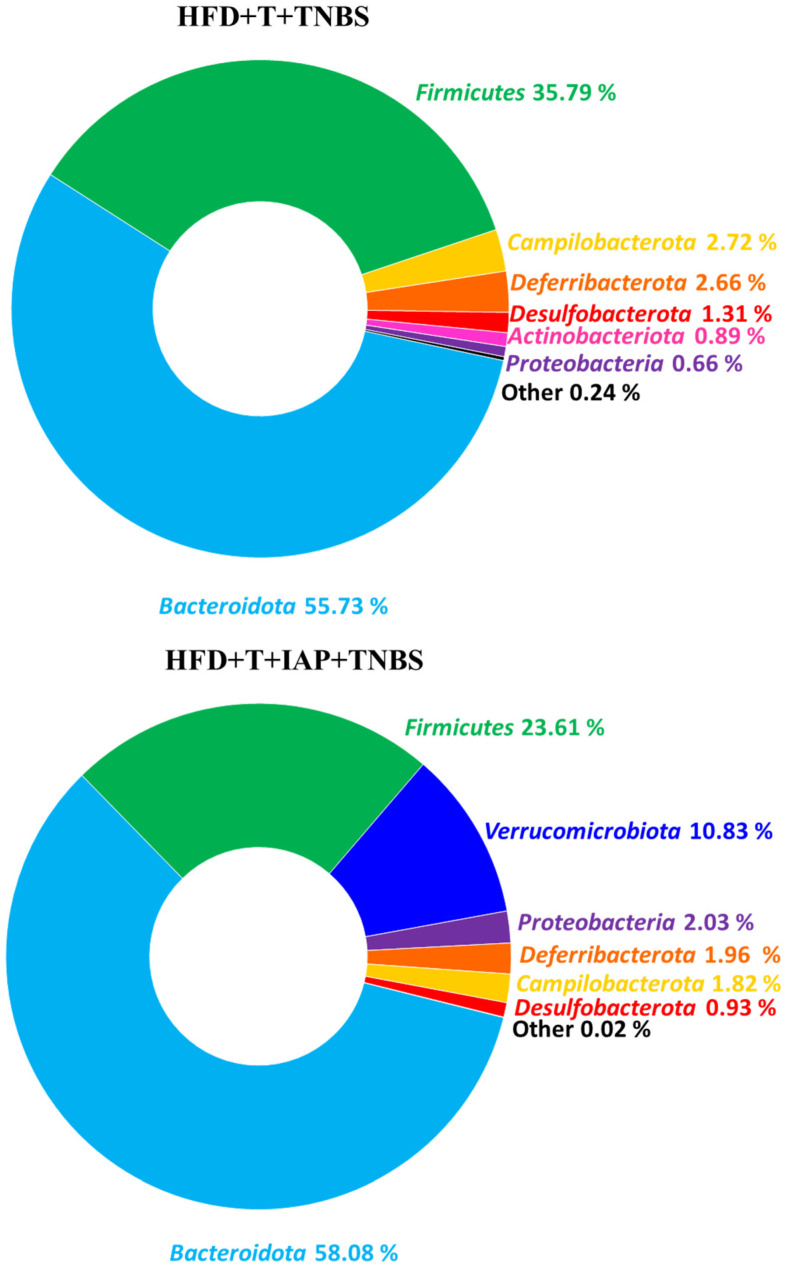
Relative abundance (%) of the gut microbiome at the phylum level for obese, treadmill-exercising mice with TNBS colitis and administered with IAP.

**Figure 7 ijms-25-00703-f007:**
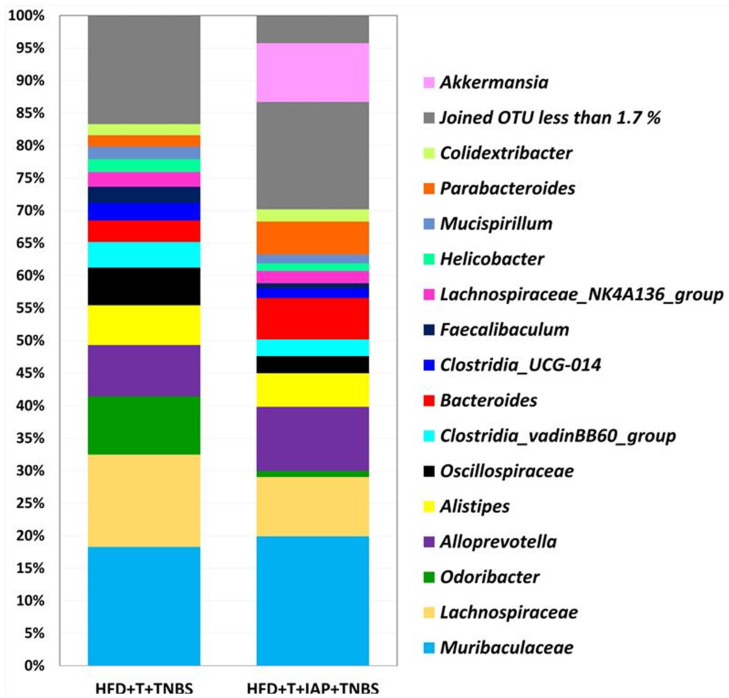
Relative abundance (%) of the gut microbiome at the genera level for obese, treadmill-exercising mice administered with IAP and with induced colitis.

**Figure 8 ijms-25-00703-f008:**
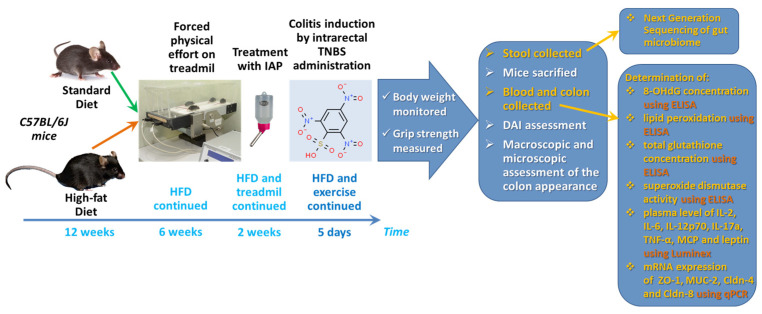
A flow chart presenting the experimental protocol, the time duration and the assessments determined in this study.

**Table 1 ijms-25-00703-t001:** Primer sequences and annealing temperature of PCR products.

PCR Product	Primer Sequence	Product Size (bp)	Annealing Temperature (°C)
b-actin	Forward: 5′-CCCATCTATGAGGGTTACGC-3′ Reverse: 5′-TTTAATGTCACGCACGATTTC-3′	150	60
ZO-1	Forward: 5′-GTTGGTACGGTGCCCTGAAAGA-3′ Reverse: 5′-GCTGACAGGTAGGACAGACGAT-3′	133	60
Muc2	Forward: 5′-TCCTGACCAAGAGCGAACAC-3′ Reverse: 5′-GGGTAGGGTCACCTCCATCT-3′	182	60
Cldn4	Forward: 5′-TTTTGTGGTCACCGACTTTG-3′ Reverse: 5′-TGTAGTCCCATAGACGCCATC-3′	60	60
Cldn8	Forward: 5′-GGGCCTGGGGATAAAAGAG-3′ Reverse: 5′-AATCCTTAAGCTGTTTTTAGGCAAT-3′	60	60

## Data Availability

The data that support the findings of this study are available from the corresponding author upon reasonable request.

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
