# Peer review of "Alkaline Phosphatase Relieves Colitis in Obese Mice Subjected to Forced Exercise via Its Anti-Inflammatory and Intestinal Microbiota-Shaping Properties"

_ijms, 2024, doi:10.3390/ijms25020703_

Round 1

Reviewer 1 Report

Comments and Suggestions for Authors

This article investigated the effect of IAP treatment on experimental colitis in mice subjected to forced exercise on a high-fat diet. There are some comments given to the authors as follows:

1.Figure 3A does not have a scale as a reference. Please provide the original image of figure 3.

2. Please harmonize the measurement criteria for Figure 3B, C and place them in the same position.

3. The presentation of next-generation sequencing results is too simple and the role of differential bacteria should be further analyzed.

4. Figure 1 should not be placed last in the order.

5. The background section needs to be made more logical, and should be combined with more clinical studies to reveal the clinical implications.

6. The picture legends are not labelled with statistical methods.

Comments on the Quality of English Language

Need for extensive English editing.

Author Response

Point-by-point Response to Reviewer #1 Comments

We greatly appreciate the thorough review and insightful comments provided by this Reviewer. All these comments have been carefully considered, and our manuscript has been revised in accordance with this Reviewer's suggestions and comments. We attach our detailed letter response point-by-point. The Reviewer's comments are repeated in italics, followed by us as authors (AU) response and the actions taken.

Rev#1: Figure 3A does not have a scale as a reference. Please provide the original image of figure 3. 

AU: We appreciate this comment raised by Reviewer. The original image of Figure 3A has changed its configuration and it has been provided with an appropriate scale for reference as requested by this Reviewer.

Rev#1: Please harmonize the measurement criteria for Figure 3B, C and place them in the same position. 

AU: Thank you for this suggestion. We have harmonized the measurement criteria for Figures 3B and 3C and placed them in the same position.

Rev#1: The presentation of next-generation sequencing results is too simple, and the role of differential bacteria should be further analyzed. 

AU: We fully agree with the Reviewer's comment that a more detailed analysis of the microbiota would be useful. We were unable to perform additional analysis due to lack of funding associated with his project, but in response to a Reviewer's comment, we have expanded the section on next-generation sequencing results and included a more detailed analysis of the role of differentiating bacteria. We will continue assessment of these interesting microbial aspects in relation to exercise and experimental colitis after raising funds in future studies.  

Rev#1: Figure 1 should not be placed last in the order. 

AU: Our credit goes to Reviewer for her/his feedback. Our manuscript was prepared according to guidelines for authors presenting work in IJMS journal, that is why Material and Methods Section was edited as the last section along with previous “methodological” Figure labelled as Figure 1. However, now we have rearranged the figures and Figure 8 is now placed as last Figure in this order.

Rev#1: The background section needs to be made more logical and should be combined with more clinical studies to reveal the clinical implications. 

AU: We appreciate the comment of this Reviewer for improvement of the background section editing. We have now revised the background section to make it more logical and have incorporated more clinical studies to highlight the practical implications as requested by this Reviewer. Moreover, the translational option of our experimental study is emphasized in our manuscript conclusions (please, see last para of Discussion Section).

Rev#1: Need for extensive English editing.

AU: Thank Reviewer for this suggestion. We have thoroughly checked and corrected the entire manuscript with the help of a native English speaker.  Again, we express our sincere thanks for this Reviewer#1 for thorough review of our paper and we believe that these Reviewer’s comments and suggestions have significantly improved the quality of our manuscript.  

Reviewer 2 Report

Comments and Suggestions for Authors

According to the manuscript by Dagmara Wojcik-Grzybek and colleagues entitled "Alkaline phosphatase relieves colitis in obese mice subjected to forced exercise via its anti-inflammatory and intestinal microbiota-shaping properties". Intestinal alkaline phosphatase (IAP) is an enzyme that protects the digestive system. As part of this study, mice subjected to forced exercise on a high-fat diet were treated with IAP to determine the effect of this treatment on experimental colitis.We fed C57BL/6 mice with TNBS colitis a high-fat diet and subjected them to forced treadmill exercise, either with or without IAP. We assessed disease activity, oxidative stress, inflammatory cytokines, and gut microbiota. Exercise exacerbated colitis in obese mice, as evidenced by an increase in disease activity index (DAI), oxidative stress markers, and proinflammatory adipokines and cytokines. In the colonic mucosa, IAP treatment reduced these effects and promoted the expression of barrier proteins. Additionally, IAP treatment altered the composition of the gut microbiota, favoring beneficial Verrucomicrobiota and reducing pathogenic Clostridia and Odoribacter. Based on our findings, we conclude that IAP treatment reduces the worsening effect of forced exercise on murine colitis by attenuating oxidative stress, downregulating proinflammatory biomarkers, and modulating the gut microbiota. There is a need for further investigation of IAP as a potential therapeutic strategy for ulcerative colitis. Regarding the present manuscript, I would like to make a few comments.

  • This manuscript appears to be a continuation of an earlier work with the aim of raising the limitations of the previous work

  • In the introduction section, the theme is explained in a clear and concise manner.

  • In the results section, it is the figures that are the major concern, both in terms of quality and solution. Does the author have the option of putting the figures 1 and 3 in color rather than gray?

  • According to the manuscript structure, the material and methods are in the last section, please renumber Figure 1.

  • Considering the large number of variables measured by the authors, what is the impact of intestinal alkaline phosphatase?

  • Microbiota (16S) requires more attention. Some references are out of date and the process needs to be explained

  • While I enjoyed reading the manuscript, I believe that an integrated presentation of all the data is lacking

Author Response

Point-by-point Response to Reviewer #2 Comments

We appreciate the careful review and insightful comments from this Reviewer. All these critical comments and kind suggestions raised by this Reviewer have been considered in revised version of our manuscript accordingly. Here is our point-by-point response. The comments from the Reviewer are reiterated in italics, followed by us as authors response (AU) and the actions taken.

Rev#2: This manuscript appears to be a continuation of an earlier work with the aim of raising the limitations of the previous work. 

AU: We acknowledge Reviewer comment. Indeed, this manuscript is a continuation of our previous work, aiming to address previous limitations and addressing broader aspects of exercise including analysis of microbiota, oxidative stress biomarkers and expression of intestinal barrier proteins.

Rev#2: In the introduction section, the theme is explained in a clear and concise manner.

AU: We express our gratitude to this Reviewer for this favourable comment. We added more recent clinical evidence in this field as requested by another Reviewer #1.

Rev#2: In the results section, it is the figures that are the major concern, both in terms of quality and solution. Does the author have the option of putting the figures 1 and 3 in colour rather than grey? 

AU: Thank you for your suggestion. We have revised Figures 1 and 3 and both columns for data referring to obese animals are now presented in green colour instead of grey colour left in these Figures to mice fed standard diet.

Rev#2: According to the manuscript structure, the material and methods are in the last section, please renumber Figure 1. 

AU: Our credit goes to Reviewer for her/his feedback. Our manuscript was prepared according to guidelines for authors presenting work in IJMS journal, which is why Material and Methods Section was edited as the last section along with previous “methodological” Figure 1. However, now we have renumbered Figure 8 (previous Figure 1) in accordance with the useful comment of this Reviewer.

Rev#2: Considering the large number of variables measured by the authors, what is the impact of intestinal alkaline phosphatase? 

AU: Again, Reviewer has addressed an important issue. Our study shows that intestinal alkaline phosphatase (IAP) can alleviate the course of experimental colitis in obese mice. We have provided an insight into this beneficial effect of IAP by demonstration that exogenous administration this protective enzyme reduced pro-inflammatory biomarkers, attenuated biomarkers of oxidative stress, improved the expression of intestinal barrier tight junction proteins, and altered composition and diversity of intestinal microbiota and this conclusion was incorporated into the Abstract (in limited words) and last paragraph of Discussion.

Rev#2: Microbiota (16S) requires more attention. Some references are out of date and the process needs to be explained. 

AU: We thank Reviewer #2 you for pointing the determination and significance of microbiota in our manuscript. We have updated the references and provided a more detailed explanation of the process of examination of microbiota (16S).

Rev#2: While I enjoyed reading the manuscript, I believe that an integrated presentation of all the data is lacking. 

AU: We appreciate Rev #2 feedback concerning general impression on our manuscript and her/his opinion on integration of all our data. We have made our best effort in revised version of our manuscript to present all the data in a more integrated manner.

Again, we would like to express our sincere gratitude to Reviewer #2 for the critical comments that all were considered in the revised version of our manuscript, and we believe that these suggestions have significantly improved the quality of our manuscript.